



# Experimental analysis of the effect of dynamic induction control on a wind turbine wake

Daan van der Hoek[1], Joeri Frederik[1], Ming Huang[2], Fulvio Scarano[2], Carlos Simao Ferreira[2], and Jan-Willem van Wingerden[1]

[1]Delft Center for Systems and Control, Faculty of Mechanical, Maritime and Materials Engineering (3mE), Delft University of Technology, Delft, The Netherlands
[2]Department of Aerodynamics, Wind Energy, Flight Performance and Propulsion, Faculty of Aerospace Engineering, Delft University of Technology, Delft, The Netherlands

**Correspondence:** Daan van der Hoek (d.c.vanderhoek@tudelft.nl)

**Abstract.** Dynamic induction control (DIC) has proven to be an effective method of increasing the power output for a wind farm in both simulation studies and wind tunnel experiments. By pitching the blades of a wind turbine periodically, the recovery of the low velocity wake is accelerated, thereby increasing the energy available to downstream turbines. The wake itself of a turbine operating with DIC has not yet been studied experimentally. This paper presents a wind tunnel experiment where

the wake of a wind turbine under periodic excitation is investigated. Using three-dimensional particle image velocimetry, the velocity field behind the turbine was reconstructed. Analysis of the velocity fields indicated that the available power in the wake increases when using DIC. This increase was partially due to a lower average thrust force experienced by the turbine with DIC. However, a large difference was seen between measurement results and actuator disk theory, indicating enhanced recovery of the wake is contributing to the increased energy. Instantaneous measurements visualizing the development of blade

tip vortices also showed how the location of vortex breakdown, which is directly related to re-energizing the wake, shifts over time with DIC. We believe this shifting location is contributing to the enhanced wake recovery of DIC, providing more energy to downstream wind turbines.

## 1 Introduction

The area that is covered by (offshore) wind farms is often limited by technical or economic reasons, resulting in closely spaced

wind turbines. While placing turbines close to each other leads to lower installation and maintenance costs, it also introduces energy losses due to the aerodynamic interaction between turbines. As a turbine extracts energy from the wind, a slower and more turbulent airflow is left in its wake. When multiple turbines are aligned with the incoming wind direction, they will experience the wake from upstream turbines. Current industry practice is to operate each wind turbine individually, so-called greedy control, thereby disregarding the impact from wakes on nearby turbines. This results in sub-optimal power production

and increased structural loading of downstream turbines (Barthelmie et al., 2010; Kanev and Savenije, 2015).

Wake losses in wind farms can be partially mitigated by actively manipulating the wakes of upstream turbines using wind farm control (Boersma et al., 2017). Wind farm control methods can generally be categorized into wake redirection control and





axial induction control (Kheirabadi and Nagamune, 2019). Wake redirection control usually refers to purposefully misaligning a turbine with the wind (e.g., Kanev et al., 2018). This changes the direction of the force applied by the turbine on the incoming airflow, thus deflecting the wake away from downstream turbines. While the misaligned turbines produce less power, the overall power generated by a farm increases (e.g., Gebraad et al., 2016; Campagnolo et al., 2016; Fleming et al., 2017, 2019). Axial induction control distinguishes between static and dynamic methods. With static induction control (SIC), the upstream turbine is derated in order to extract less energy from the wind, leaving more for downstream turbines (Kanev et al., 2018). Although steady-state wake models indicate a potential increase in power production using SIC (Annoni et al., 2016), high-fidelity simulations and field tests have shown either limited or no benefits (Annoni et al., 2016; Campagnolo et al., 2016; van der Hoek et al., 2019; Frederik et al., 2020). More promising results have been obtained using dynamic induction control (DIC), where the induction factor of a turbine is varied over time. This action is thought to induce additional turbulence in the wake of a turbine, which enhances mixing and subsequently results in faster wake recovery (Frederik et al., 2020).

High-fidelity simulations have shown that it is possible to improve the power production of a wind farm by optimizing the control inputs for dynamic induction (Goit and Meyers, 2015; Munters and Meyers, 2017). However, finding the optimal excitation sequence is very computationally intensive. Goit and Meyers (2015) first applied this approach in large eddy simulations. The optimal thrust coefficient of the turbines over time was determined using the adjoint-based gradient, and implemented using a receding horizon approach. This turned out to be highly successful, increasing energy extraction by 16%. However, this method has a number of features that make it impossible to implement directly in a real-world environment. First, the objective function used in the optimization sequence requires perfect knowledge of the flow, which is not available in real life. Second, the computational time necessary for the gradient calculation was substantially larger than the receding horizon time window, which makes real-time implementation infeasible. Third, the authors controlled the turbine's thrust coefficient directly, whereas in practice it depends on multiple parameters, both internal (e.g., pitch angle, tip-speed ratio) and external (e.g., wind speed, turbulence).

Munters and Meyers (2018) addressed the first two challenges by introducing periodic thrust variations. Although this approach reduces the potential power gain slightly, it is more practical to implement. As the optimization is now limited to finding the optimal thrust amplitude and frequency, the computationally expensive optimization sequence is no longer needed. The final challenge is to translate the thrust coefficient variation into a control signal for the wind turbine. Frederik et al. (2020) proposed a straightforward approach where a periodic variation of the pitch angle is superimposed to the baseline controller. While this approach slightly alters the applied thrust variation, wind tunnel experiments with a three turbine array showed a similar wake mixing effect, leading to a 2–4% gain in power (Frederik et al., 2020).

The experiments by Frederik et al. (2020) demonstrated the potential benefit of DIC, but they only considered power measurements taken directly from the wind turbines. No experiments have been conducted thus far that characterize the aerodynamic behaviour of a wind turbine wake in DIC regime, which is vital to gain a better understanding of its physical working principle. The goal of this paper is to investigate the effect of dynamic induction control on a wind turbine wake. To do this, the wake of a small-scale wind turbine model is measured using three-dimensional particle image velocimetry (PIV), a technique





used successfully in wind tunnel experiments to characterize the wake of scaled wind turbine models (e.g., Rockel et al., 2014; Lignarolo et al., 2014, among others).

The paper is organized as follows. Section 2 presents the setup used for carrying out the experiments, which includes the turbine model, control strategy, and PIV system. In Sect. 3, the measurement results are shown in the form of flow fields. Some of the key measurement results are discussed further in Sect. 4. Finally, conclusions on the performance of DIC are given in Sect. 5.

## 2 Experimental setup

First, we present the wind tunnel and the scaled wind turbine model. Next, the wind turbine control methods used during the experiment are discussed. Finally, the PIV setup used to measure the wind turbine wake is introduced.

### 2.1 Wind tunnel and turbine model

The experiments were done in the Open Jet Facility (OJF) of the Aerospace Engineering Faculty at TU Delft. This is a closed-loop open jet wind tunnel with an octagonal outlet of $2.85\,\mathrm{m}$ by $2.85\,\mathrm{m}$. The wind tunnel has a contraction ratio of 3:1, and can reach a free stream velocity of $35\,\mathrm{m\,s^{-1}}$ with a turbulence intensity (TI) between 0.5 and 2 % (Lignarolo et al., 2015). The current experiments were conducted at a wind speed of $4\,\mathrm{m\,s^{-1}}$.

The turbine model used for the experiments was the MoWiTO-0.6 developed at the University of Oldenburg (Schottler et al., 2016). This turbine has a rotor diameter of $D = 0.58\,\mathrm{m}$, and is equipped with a generator that allows torque control to regulate the rotor speed. A stepper motor attached to a slider is used to pitch the blades collectively. An overview of the model including relevant dimensions is provided in Fig. 1. To measure the deformation of the tower, strain gauges were applied to its base on the front and back in a full Wheatstone bridge configuration. These strain measurements were used to reconstruct the thrust force acting on the rotor. See Schottler et al. (2016) for additional information on the design of the wind turbine.

Communication between wind turbine and computer was arranged with a *dSPACE MicroLabBox*. The frequency for communication was set to $10\,\mathrm{kHz}$, while measurements were recorded with a frequency of $1\,\mathrm{kHz}$. Recorded turbine signals included the rotor speed, pitch angle, generator current and strain at the tower base.

The steady-state aerodynamic coefficients, electrical power $C_{P_e}$ and thrust $C_T$, were determined experimentally at the start of the measurement campaign for different tip-speed ratios and pitch angles. The electrical power $P_e$ could be reconstructed from the generator current using a generator specific constant (Schottler et al., 2016). The thrust force $T$ was acquired by measuring the strain at the tower base, and relating this to a calibration function. The measurements indicated that the turbine performed optimally for a tip-speed ratio of $\lambda_* = \Omega R/U = 5.5$, with rotor speed $\Omega$, rotor radius $R$ and wind speed $U$. The coefficients $C_{P_e}$ and $C_T$ for this tip-speed ratio are given in Fig. 2 for a range of pitch angles. The 0° pitch angle was defined as the most outward position that the blades could reach. For a blade pitch angle of $\beta_* = 9°$ optimal power was achieved, while simultaneously guaranteeing a reasonable thrust coefficient.



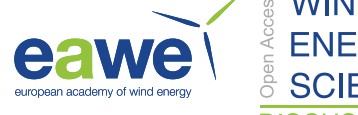

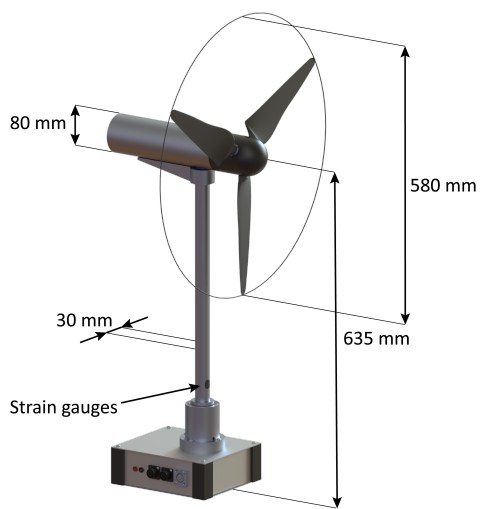

**Figure 1.** MoWiTO-0.6 wind turbine model including relevant dimensions.

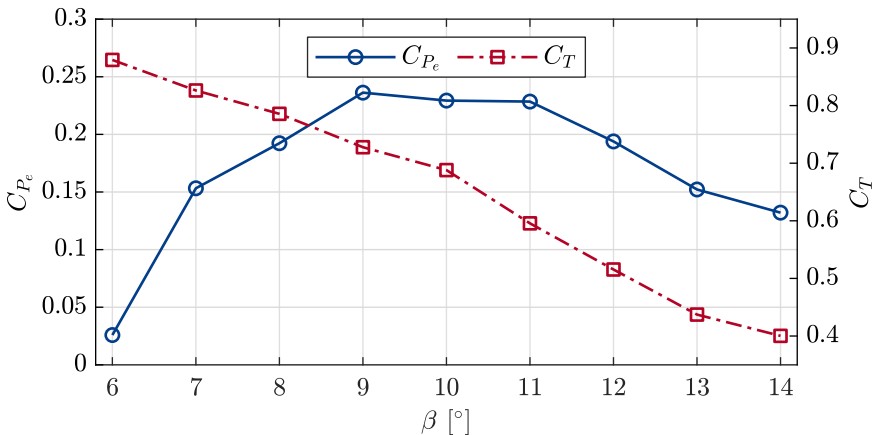

**Figure 2.** Experimentally determined power ($C_{P_e}$) and thrust ($C_T$) coefficients of the MoWiTO-0.6 for different pitch angles $\beta$ at the optimal tip-speed ratio $\lambda_* = 5.5$.

## 2.2 Dynamic induction control strategy

The wind turbine was operated with two different control methods. First, we used the classical *greedy* control approach, where
a turbine maximizes its own power output without considering interaction with other turbines. This control strategy served as
a baseline for the second method, which was dynamic induction control.





For the baseline case, we implemented the simple *K-omega-squared* control strategy from (Bossanyi, 2000) to regulate the rotor speed. In this case, a generator torque is applied on the turbine according to the control law

$$Q_e = K\Omega^2, \tag{1}$$

where $Qe$ is the electrical torque. The desired torque was applied by the generator, which was in turn controlled by a shunt resistor. The controller gain $K$ was based on the following relation for the optimal operating settings:

$$K = \frac{\rho_{\mathrm{air}}\pi R^5 C_{P_e}(\lambda_*,\beta_*)}{2\lambda_*^3}, \tag{2}$$

with $\rho_{\mathrm{air}}$ indicating air density and $R$ the rotor radius. Using this control law, the turbine should be operating at the top of its power coefficient curve (see Fig. 2). The blade pitch was kept constant at the optimal value.

A similar controller to the one used by Frederik et al. (2020) was implemented for dynamic induction control. This controller also used the control law from Eq. (1) to regulate the rotor speed. Additionally, by imposing a sinusoidal signal on the collective blade pitch angles, the thrust coefficient of the turbine could be varied over time. The amplitude and frequency of these variations were limited by the bandwidth of the pitch motor. Previous studies have indicated that the optimal pitch frequency is found for a Strouhal number in the range of $St = 0.25$ (Munters and Meyers, 2018; Frederik et al., 2020), where the Strouhal

number is defined as

$$St = \frac{fD}{U}, \tag{3}$$

with $f$ the excitation frequency. Note that the Strouhal definition implies that the excitation frequency scales with the turbine rotor diameter. For large wind turbines with rotor diameters of up to $180\,\mathrm{m}$, this optimal Strouhal number results in a very low frequency (typically $\ll 0.1\,\mathrm{Hz}$). The corresponding pitch frequency for the MoWiTo-0.6 model is equal to $f = 1.72\,\mathrm{Hz}$. Initial

testing with the turbine showed a maximum pitching speed around $16\,°\mathrm{s}^{-1}$, limiting the pitch amplitude to $2.3°$ at $St = 0.25$. To ensure that a range of different frequencies around this reported optimal frequency could be tested, we varied the blade pitch angle between $8$–$12°$. Note that this variation was not centered around the optimal pitch angle of $\beta = 9°$. This was done to ensure a higher thrust variation and smaller loss of power according to the aerodynamic coefficients from Fig. 2.

  The periodic pitch signal resulted in a varying thrust coefficient with values between 0.52–0.79, which was a slightly lower

amplitude than the cases presented in Frederik et al. (2020). We expect smaller wake mixing effects due to the smaller variation in thrust. However, Frederik et al. (2020) conducted their experiments at TIs of 5 and 10 %, whereas the experiments presented here were conducted at a lower TI. A higher TI results in a higher amount of natural wake mixing, regardless of the wind turbine control action. Therefore, the low TI conditions ensured that only a small pitch action was needed to observe significant perturbations in the wake.

The evolution of the thrust coefficient over time with this sinusoidal pitch reference is depicted in Fig. 3. This figure is based on the steady-state thrust coefficient from Fig. 2, and merely illustrates the varying thrust signal. During the experiment we expect slightly altered signals, since the thrust force is affected by dynamic inflow. Compared to the baseline case, a thrust coefficients above 0.72 is considered *over-inductive*: the induction is higher than the optimal in terms of steady-state energy





capture. A similar variation from over-inductive to under-inductive thrust coefficients was used in previous research (Munters
and Meyers, 2018; Frederik et al., 2020). It is expected that these variations lead to increased turbulence in the wake and,
subsequently, faster wake recovery.

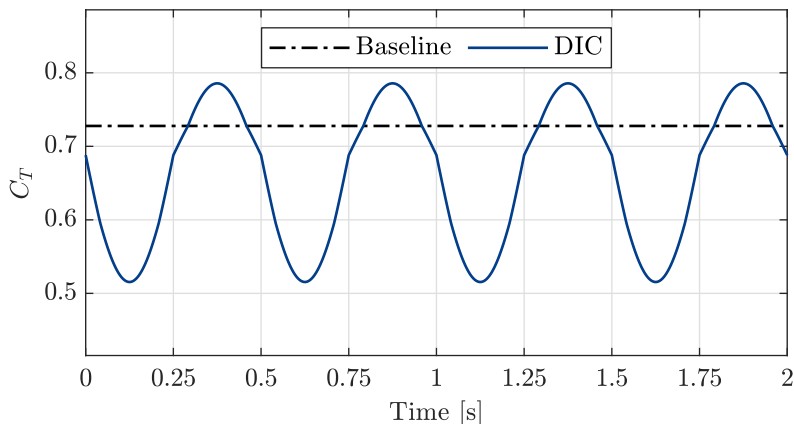

**Figure 3.** Thrust coefficient reference for baseline control and periodic dynamic induction with a frequency of 2 Hz ($St = 0.29$).

The wind tunnel experiments from Frederik et al. (2020) did not show a clear optimum over the Strouhal numbers. However,
small differences in power gain were observed for different frequencies. In our experiment, we applied DIC at Strouhal numbers
of 0.19, 0.24 and 0.29, deeming these sufficient for including the optimal Strouhal number with this particular setup.

### 130    2.3    Three-dimensional velocity measurements

The wake of the turbine was measured using a large-scale PIV system. An overview of the setup is shown in Fig. 4. Helium-
filled soap bubbles (HFSBs) were used as flow tracers (Scarano et al., 2015). An aerodynamic seeding rake at the outlet of
the open jet released a continuous stream of particles into the incoming flow. The seeder introduced additional disturbances
to the flow, but its position was deemed necessary for sufficient seeding of the measurement volume. Three *LaVision LED*
*Flashlights* were used to illuminate the particles in the turbine wake. A setup consisting of three *Photron FASTCAM SA1.1*
high-speed cameras recorded the light scattered by the seeding particles at a frequency of 500 Hz, with a resolution of 1024
by 1024 pixels. The field of view (FOV) covered by the cameras was approximately $0.75\,\mathrm{m} \times 0.5\,\mathrm{m} \times 0.75\,\mathrm{m}$. A *High Speed
Controller* from *LaVision* was used to synchronize the illumination and imaging. Each recording lasted $10\,\mathrm{s}$, resulting in 5000
images for each measurement.

Processing the recorded images consisted of several steps. First, a high-pass filter was applied to the raw images in order to
remove any undesired background light (Sciacchitano and Scarano, 2014). The filtered images were subsequently processed
with the *DaVis 10* software from *LaVision*. This process included a self-calibration using recorded images to correct any
misalignments between cameras and the measurement volume (Wieneke, 2008). Next, the motion of the tracers was obtained
by tracking them individually (particle tracking velocimetry) using the efficient Shake-The-Box (STB) algorithm from Schanz





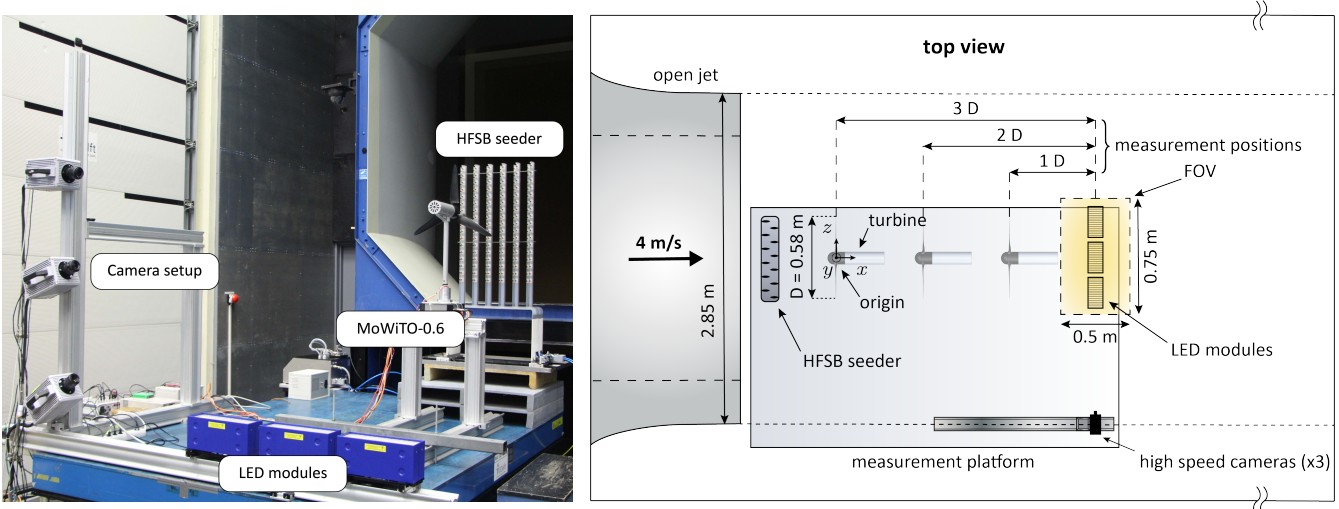

**Figure 4.** Experimental setup for measuring the wake using particle image velocimetry (left) and a schematic overview of the setup highlighting the different components and turbine locations used for measuring the wake (right).

et al. (2016). This algorithm reconstructs the particles' trajectories from the recorded data based on the principle of Lagrangian particle tracking (LPT), providing the velocity vector along trajectories in a three-dimensional domain. Finally, the scattered data from the particle positions and velocities was reduced onto a Cartesian grid by spatially averaging over small cubic cells (binning). Time-averaged velocity fields were rendered using a bin size of $50\,\mathrm{mm} \times 50\,\mathrm{mm} \times 50\,\mathrm{mm}$. A 75 % overlap among neighboring bins yielded a final vector spacing of $12.5\,\mathrm{mm}$. For instantaneous velocity fields, the bin size was increased by 50 % and a temporal moving-average filter with a window of $0.01\,\mathrm{s}$ was applied in order to collect sufficient particles for the binning procedure.

By moving the turbine to three different positions, as indicated in Fig. 4, multiple regions along the wake were measured. The distance between the rotor plane and the center of the FOV was varied between one and three diameters $D$ downstream, obtaining an effective measuring range of 0.5–3.5 $D$. Repositioning the turbine instead of the measurement system was the strategy adopted to reduce the time needed for experiments, although the conditions upstream of the turbine were affected by a varying distance from the HFSB seeding rake.

## 3   Results

This section presents the results obtained from the measurements described in the previous section. Section 3.1 contains the thrust coefficient measured at the turbine. In Sect. 3.2, we show the time-averaged flow fields at multiple distances behind the turbine. The velocity fields are averaged over the DIC pitch cycle in Sect. 3.3, providing a measure of the available power in the wake over time. Finally, instantaneous measurements are used to visualize the development of the blade tip vortices in Sect. 3.4.



### 3.1 Turbine thrust coefficient

The performance of the turbine at the three measurement positions was compared using the thrust force measured by strain

gauges at the tower base. The measured thrust forces were transformed to thrust coefficients using

$$C_T = \frac{T}{\frac{1}{2}\rho\pi R^2 U_\infty^2},\tag{4}$$

with thrust force $T$, air density $\rho$, and rotor radius $R$. A constant inflow velocity of $U_\infty = 4\,\mathrm{m\,s^{-1}}$ was used for the computation. Figure 5 shows $C_T$ for the baseline case and one of the DIC cases ($St = 0.24$). We observe that the average $C_T$ in both cases (baseline and DIC) was up to $20\,\%$ lower compared to the expected values from Fig. 3, which were based on the steady-

state thrust coefficient. Furthermore, we see that the difference in average thrust coefficient between baseline and DIC varied depending on the location of the turbine. The average $C_T$ decreased for the baseline case, whereas it increased for the DIC case.

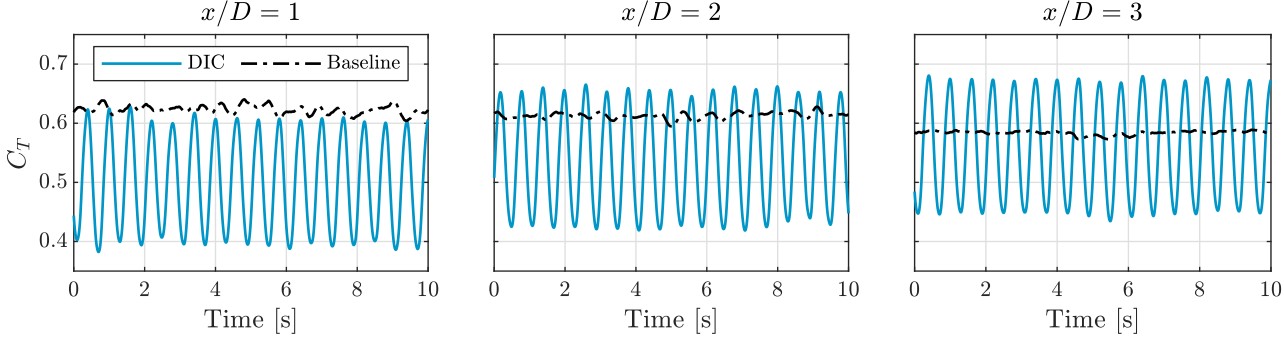

**Figure 5.** Thrust coefficient of the turbine for different measurement positions. The results are based on a single measurement for each distance using either baseline control or DIC with $St = 0.24$.

We believe there are several explanations for the variations in the measured $C_T$. First, the HFSB seeding rake was not present when the aerodynamic coefficients from Fig. 2 were obtained. This could indicate that the seeding rake affected the incoming

flow velocity $U_\infty$. Furthermore, we expect that the influence of the seeding rake changes depending on how close the turbine is positioned from it. Second, the relatively low wind speed used for the experiments introduced some unsteadiness in the inflow, and was also more susceptible to measurement errors in the wind speed at the outlet of the open jet. Finally, the turbine controller was designed using the aerodynamic coefficients that were obtained without the seeding rake. In the DIC case, the rotor speed varied periodically due to the pitch action, making it more difficult for the controller to operate the turbine at the

optimal tip-speed ratio.

The measurements shown in Fig. 5 make it clear that the flow fields from different turbine positions should be evaluated carefully. Due to the relative difference in $C_T$ between the baseline case and DIC, we are basically comparing different cases




at each turbine location. However, we believe the principles behind dynamic induction control can still be investigated through these measurements.

## 3.2 Time-averaged flow fields

Time-averaged flow fields were obtained from the binning procedure described in Sect. 2.3, using all the particle data from each measurement. Contours of the stream-wise velocity field at multiple cross-sections downstream are presented in Fig. 6. A clear difference in the velocity deficit was observed between baseline control and DIC with $St = 0.24$. The results of the other pitch frequencies are not shown, as the relative differences in average velocity were minor. The coordinate system used in these figures is the same as specified in Fig. 4. Due to the combination of a lower average induction factor and the periodic excitation, the average stream-wise velocity in the wake of the turbine operating with DIC was higher than in the baseline control case. This is seen for all three measurement locations.

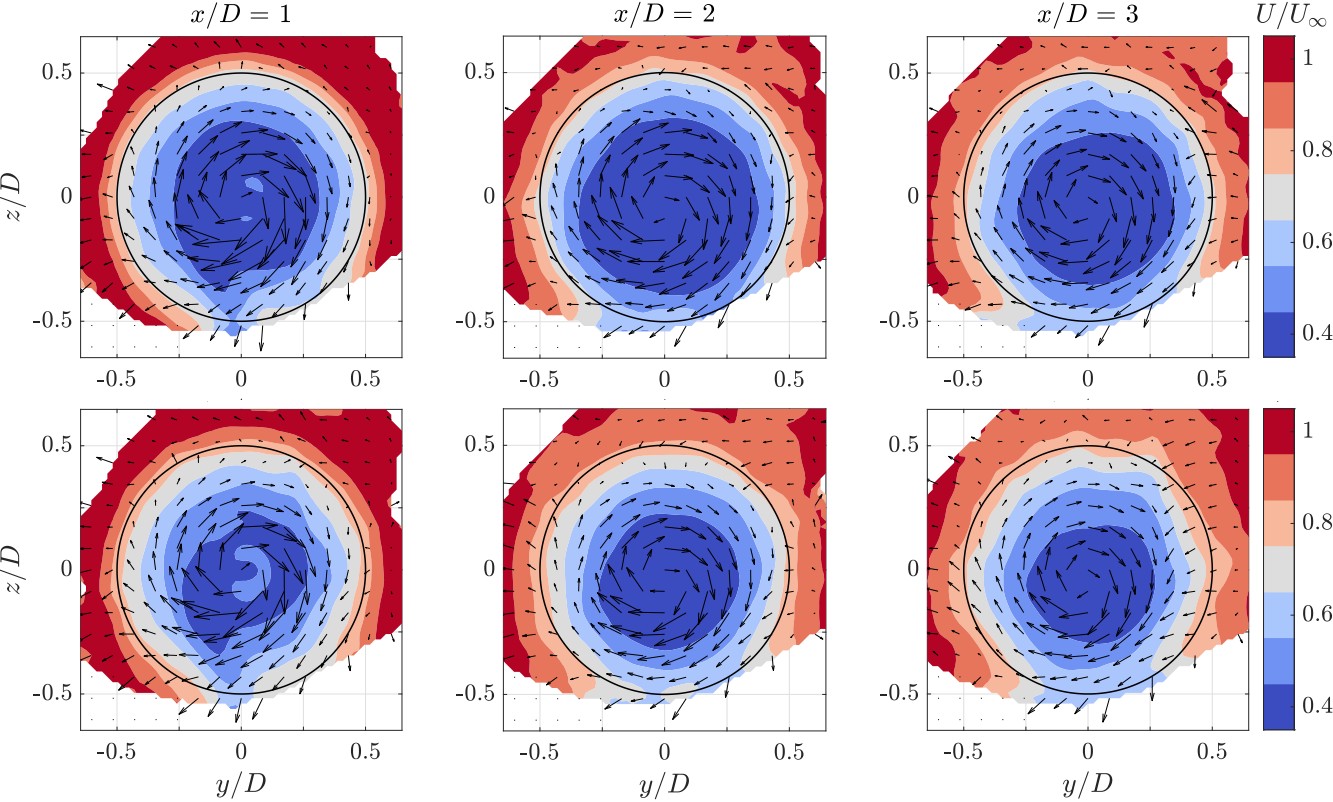

**Figure 6.** Cross-sections of the wake at distances ranging from 1 to 3 diameters downstream of a turbine operating with baseline control (top) and DIC with $St = 0.24$ (bottom). The contours show the average normalized stream-wise velocity component, while the arrows indicate the cross-stream velocity components. The black circles indicate the relative location of the rotor disk of the model wind turbine.





The measurements from different locations are merged into a single velocity field in Fig. 7. Data from the different measurement positions does not overlap, but the transitions between these measurements seem relatively smooth, even though the
turbine experienced different thrust levels. When subtracting the flow fields from the baseline and DIC case, we see clearly that there is a lower velocity deficit in most parts of the wake as a result of DIC. The increase in velocity is primarily concentrated at the edges of the wake, except at one side ($y/D < 0$) around $x/D = 3$. This asymmetry in the wake might be related to the development of the tower wake, which will be addressed later on.

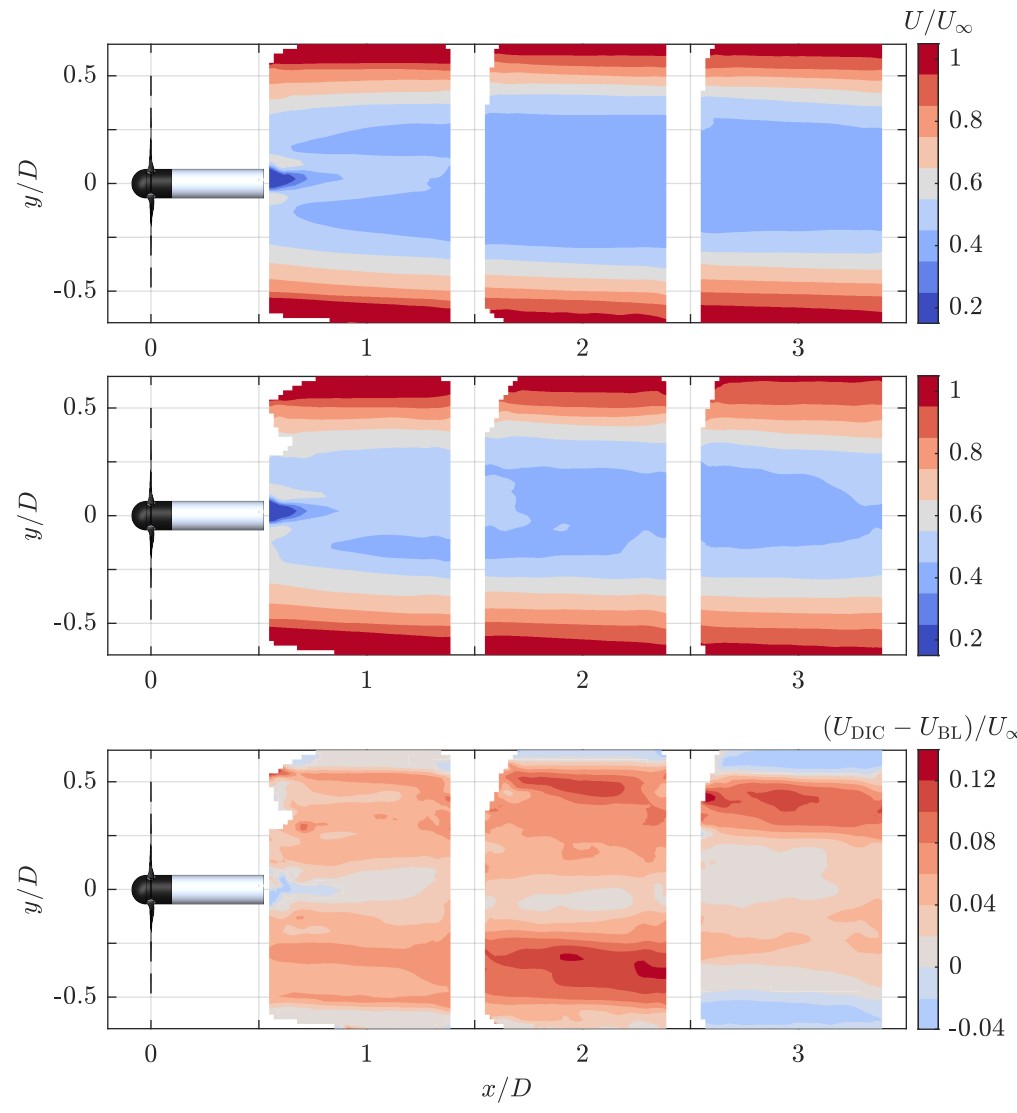

**Figure 7.** Contours of the stream-wise velocity component $U/U_\infty$ in the wake of the turbine at hub height ($z/D = 0$), operating with baseline control (top) and DIC ($St = 0.24$) (middle). The difference in velocity field between the two control methods is visualized in the bottom figure.



The time-averaged velocity fields show that the velocity, and hence the energy in the wake increased when DIC was used.
Using these flow fields, we cannot discern the contributions to the increased velocity of the periodic excitation or the lower average thrust force. However, we can get an indication of how DIC affects the recovery of the wake by inspecting the turbulence intensity. The TI at multiple locations in the wake is presented in Fig. 8. For the baseline case, we observe high TI values concentrated at the edge of the wake due to vortices released at the blade tips. The effect of these vortices spread out further downstream as the vortices started interacting with each other. Furthermore, high turbulence levels were present behind
the turbine nacelle and tower. Although the influence of the tower seemed to diminish when moving downstream, the lower left quarter of the wake shows that vortices shed by the tower started interacting with the blade tip vortices. Comparing the contours from the baseline operation with DIC, similar TI values were found at $x/D = 1$. From a distance of $x/D = 2$ and onwards, increased turbulence levels are visible in large parts of the wake, and moving towards the center.

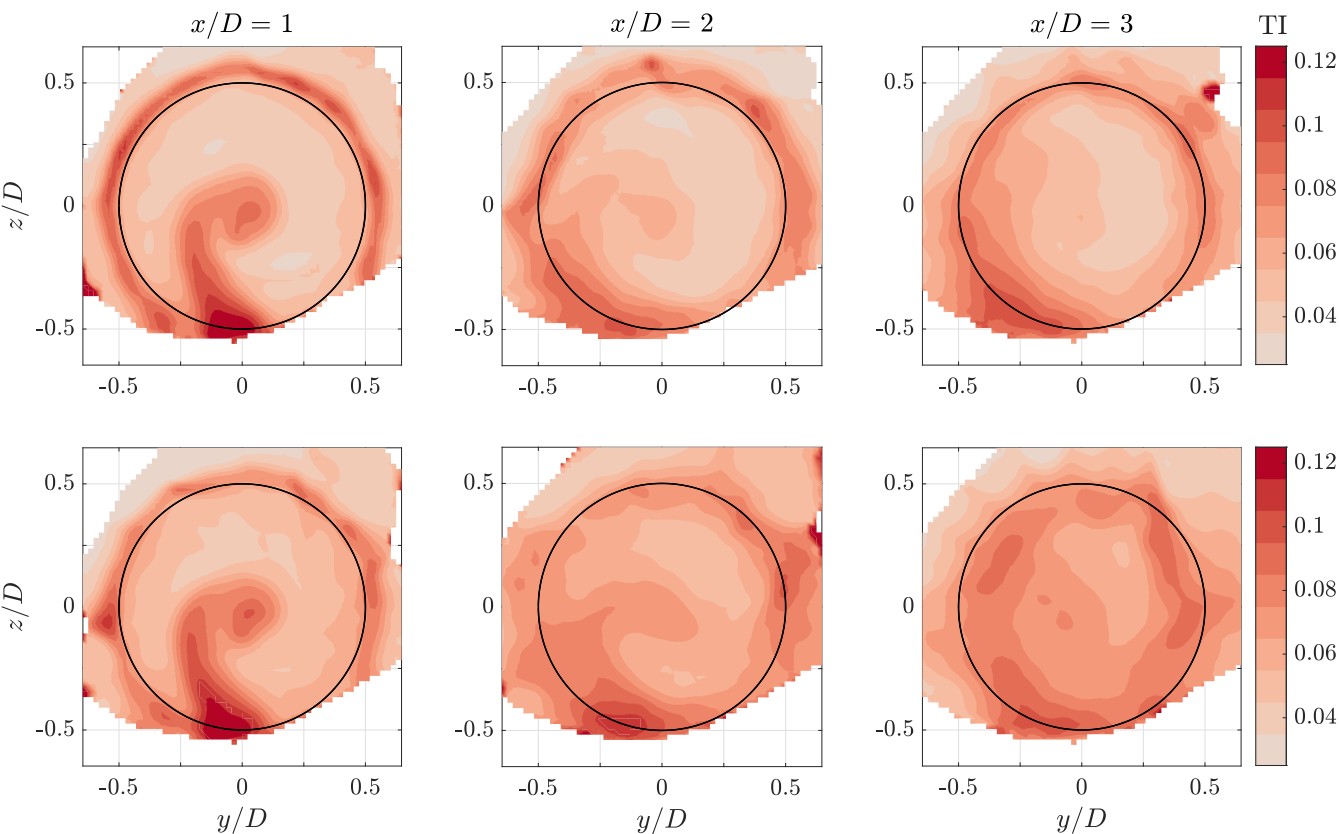

**Figure 8.** Turbulence intensity (TI) contours at multiple locations in the wake of a turbine operating with baseline control (top) and DIC with $St = 0.24$ (bottom).




## 3.3 Periodically averaged velocity and available power

This section investigates the effect of DIC on the velocity in the wake over a pitch cycle. Using the function of available power (AP) from Vollmer et al. (2016), we can quantify how the energy in the wake varied over time. The periodically averaged flow fields were obtained by dividing the instantaneous velocity fields over different phases of the pitch cycle. For a turbine pitching at a frequency of $2\,\text{Hz}$ ($St = 0.29$), this resulted in 250 measurements for a single pitch cycle (i.e., the pitch cycle was discretized in 250 phase segments). For each of these pitch phases, the data of all pitch cycles was subsequently averaged using the following equation, to obtain a higher statistical convergence:

$$U_{\text{pa}}(\beta_{\text{ph}},x,y,z) = \frac{1}{n_{\text{c}}} \sum_{\beta_{\text{ph}}} U(\beta(t),x,y,z). \tag{5}$$

Here, $U_{\text{pa}}$ is the periodic average of the velocity for a phase of the pitch cycle $\beta_{\text{ph}}$, $n_c$ represents the number of pitch cycles, and $U$ is the velocity as a function of the pitch angle $\beta$ at any time $t$. Depending on the Strouhal number, this means that the number of pitch cycles used for averaging ranged from 13–20. Phase-locked measurements are generally acquired with particle image velocimetry by timing the imaging with the principle frequency of excitation, which is usually the rotational frequency of the turbine (e.g., Lignarolo et al., 2014). However, since the pitch frequency did not always coincide with the rotor frequency, and the latter was not constant when DIC was activated, this was not a valid option for this experiment.

The result of the periodical averaging approach is shown in Fig. 9. For the baseline case, the averaging was done over a single revolution of the rotor. The resulting velocity field looks quite similar to the ones from the time-averaged figures. The velocity fields with DIC are shown at three different phases of the pitch cycle, which are indicated by the relative angle of the blades. Large differences in the velocity field over time were observed. Following a low pitch angle, corresponding to a high thrust force, the wake expanded, and a region of high velocity deficit (dark blue) was induced behind the turbine. As thrust decreased, the wake contracted while a comparatively lower velocity deficit was induced behind the turbine.

Next, the effect of DIC was evaluated using the fraction of AP as defined by Vollmer et al. (2016):

$$f_{\text{AP}}(x,r,\theta) = \int\limits_{0}^{2\pi} \int\limits_{0}^{R} U^3(x,r,\theta)/U_{\infty}^3 r \mathrm{d}r \mathrm{d}\theta. \tag{6}$$

Here, $x$ denotes the distance downstream, while $r$ and $\theta$ indicate the radial distance and azimuth angle. The equation provides the power available to a hypothetical downstream wind turbine with respect to the upstream turbine that is driven by an undisturbed flow. By applying this equation to the phase-averaged data at a distance of $x/D = 3$ downstream, the $f_{\text{AP}}$ over time is obtained in Fig. 10 for all Strouhal numbers. From the figure, it is clear that the AP downstream increased significantly for each implementation of DIC, compared to baseline operation. Small differences were observed between the different Strouhal numbers, such as the minima and maxima of the $f_{\text{AP}}$. Furthermore, it seems the pitching frequency affected the average traveling speed of the low velocity wake region, as can be seen from the time shift of the signals.

The fraction of AP was subsequently computed for the entire measurement volume and averaged over time to obtain Fig. 11. The figure indicates that the AP was initially still relatively high, due to the fact that the wake had not fully expanded

**Figure 9.** Cross-section of the periodically averaged wake at hub height ($z/D = 0$) and at three diameters downstream of a turbine operating with (a) baseline control and (b) DIC ($St = 0.24$). The contours show the normalized stream-wise velocity components and the arrows indicate the cross-stream velocity components. The grey line indicates the pitch angle at different phases of the pitch cycle.




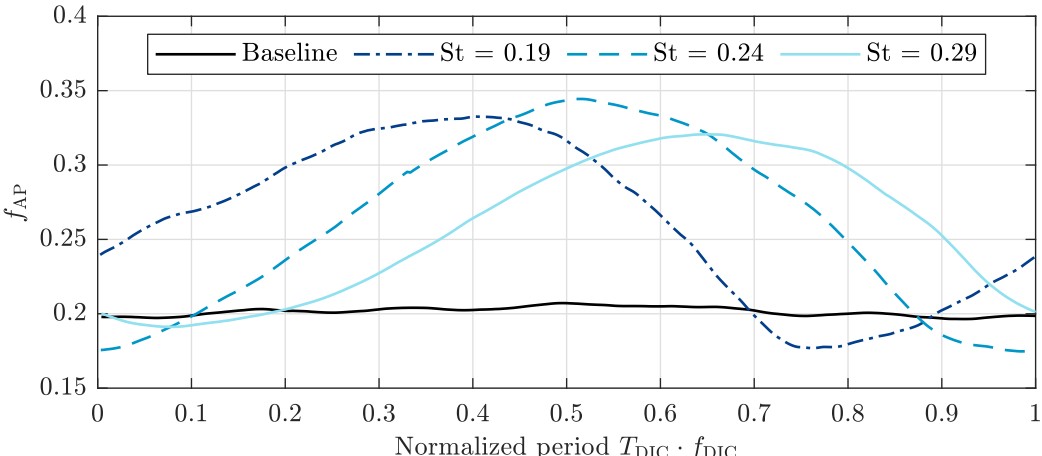

**Figure 10.** Available power $f_{\mathrm{AP}}$ over time at a distance of $x/D = 3$. The results have been normalized over time using the corresponding pitch frequencies in order to compare the results over a single pitch cycle. The different measuremens have been synchronized using the pitch reference signal.

yet. The available power reached its minimum between $x/D = 1.5$–2, after which it started to recover slowly. The different pitch frequencies for DIC showed only small differences on average; no clear optimum for the Strouhal number is observed based on the present results. The first two measurement positions, $x/D = 1$ and 2, showed a similar trend in the difference between baseline controller and DIC. However, at $x/D = 3$ we see a relative increase in the $f_{AP}$ for the baseline case. This was something that could be expected when looking at the turbine thrust measurements from Fig. 5, where the relative thrust
coefficient between baseline and DIC was a lot smaller at this measurement location.

Using the average AP from Fig. 11, we can also make a simplified calculation of the efficiency of a potential two-turbine array. By assuming that the hypothetical downstream turbine is operating with the optimal $C_{P_e}$ from Fig. 2, the power of the downstream turbine can be determined and added to the power of the upstream turbine to obtain a measure of the array power efficiency. The individual and total power coefficients for a hypothetical turbine located at $x/D = 3$ are given in Fig. 12. When
comparing the total power coefficient of the baseline case to DIC, we see that DIC was slightly less effective. This was mostly due to the amount of energy lost at the operated turbine, which was higher than in other studies (Frederik et al., 2020).

The comparison of the total power coefficient from Fig. 12 provides some insight in the effectiveness of DIC, but it relies on several assumptions. An additional evaluation can be made using one-dimensional actuator disk theory to compare the measurement results to a simple theoretical framework. In this case, the velocity in the wake of the turbine is assumed to
depend solely on the induction factor, and recovery due to turbulence is neglected. The induction factor and corresponding

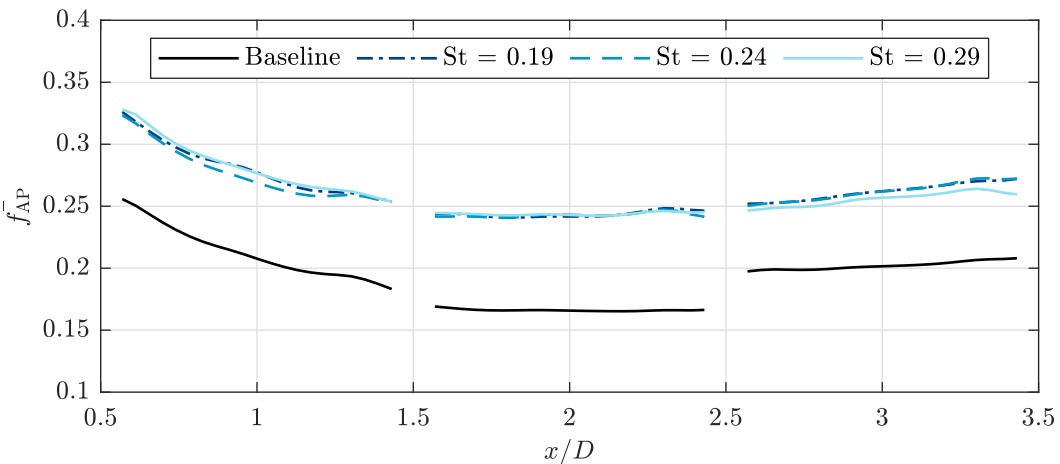

**Figure 11.** Average available power $\bar{f}_{\mathrm{AP}}$ in the wake of a turbine operated with DIC compared to baseline operation. The $\bar{f}_{\mathrm{AP}}$ is based on cross-sections of the measured stream-wise velocity field.

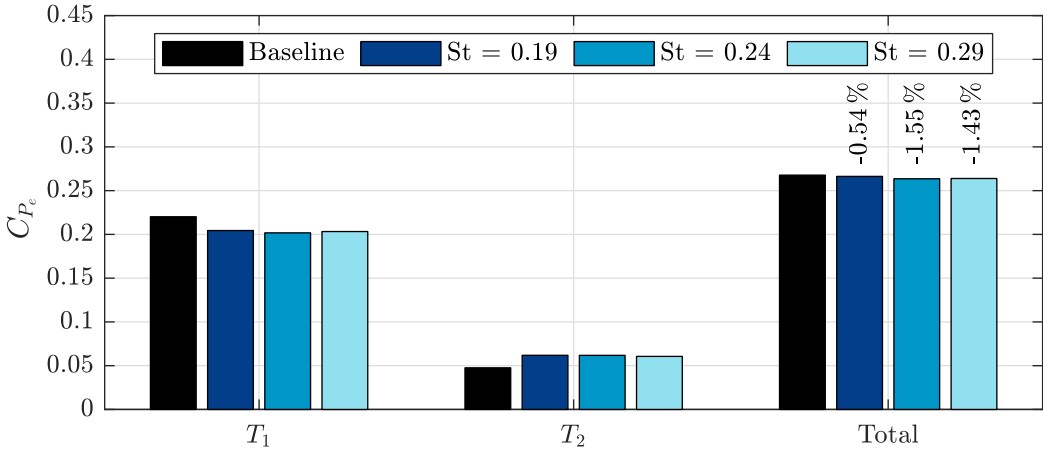

**Figure 12.** Average power coefficient $C_{P_e}$ comparing baseline control with DIC for three different frequencies. The first column shows the power of the measurement turbine, the second column shows the power of a hypothetical turbine located at a distance of $x/D = 3$ in the wake, and the final column shows the combined power of $T_1$ and $T_2$.

wake velocity are given by

$$a = \frac{1}{2}\left(1 - \sqrt{1 - C_T}\right), \tag{7}$$

$$U_w^{\mathrm{AD}} = (1 - 2a)U_\infty, \tag{8}$$





where $a$ is the induction factor and $U_w^{\mathrm{AD}}$ is the wake velocity according to actuator disk theory. For this theoretical wake
velocity we also determined the $f_{\mathrm{AP}}$ in order to compare it to the measurement results. The comparison between actuator disk
theory and the PIV measurements is provided in Table 1. The average difference between the measured thrust forces among
the various measurement cases is seen to be very small, resulting in very similar $f_{\mathrm{AP}}$ values for the actuator disk model. We
do see some differences in the amplitude of the different DIC frequencies, indicating that the pitch frequency is also affecting
the thrust amplitude. Regarding the $f_{\mathrm{AP}}$ from the PIV measurements, much larger differences are observed between baseline
and DIC. Here, a clear increase in the AP is found when DIC is being used. The difference in values between actuator disk
theory and the PIV measurements seems to suggest that a large part of the increased energy in the wake is due to enhanced
wake recovery induced by the periodic excitation from DIC.

**Table 1.** Comparison of wake properties for a hypothetical turbine at $x/D = 3$ downstream with DIC compared to baseline operation. The
different columns indicate values of the Strouhal number $St$, average thrust coefficient $\bar{C}_T$, thrust amplitude $\Delta C_T$, average induction factor
$a$, fraction of AP based on actuator disk theory $f_{\mathrm{AP}}^{\mathrm{AD}}$, and fraction of AP based on the PIV measurements $f_{\mathrm{AP}}^{\mathrm{PIV}}$.

| Control strategy | $St$ | $\bar{C}_T$ | $\Delta C_T$ | $a$ | $U_w^{AD}$ | $f_{\mathrm{AP}}^{AD}$ | $f_{\mathrm{AP}}^{\mathrm{PIV}}$ |
|---|---|---|---|---|---|---|---|
| Baseline | - | 0.58 | 0 | 0.18 | 2.57 | 0.27 | 0.20 |
| | 0.19 | 0.57 | 0.13 | 0.17 | 2.60 | 0.28 | 0.26 |
| DIC | 0.24 | 0.56 | 0.12 | 0.17 | 2.64 | 0.29 | 0.26 |
| | 0.29 | 0.56 | 0.10 | 0.17 | 2.64 | 0.29 | 0.26 |

## 3.4 Tip vortex development

The phase-averaged velocity fields allowed us to investigate some of the periodic behaviour inside the wake. However, a large
part of the dynamics is lost due to the averaging procedure. An additional study of the instantaneous data was made to find
more information on the wake mechanics of DIC. Processing the time-resolved measurements included a temporal moving
average filter with a time window of $0.1\,\mathrm{s}$. Furthermore, the bin size used for the averaging procedure was set to $75\,\mathrm{mm}$. Both
additions were made to increase the number of particles used for the spatial averaging process. Increasing the bin size any
further would have resulted in a loss of resolution where vortical structures in the wake are no longer distinguishable.

After processing the PIV measurements, the $Q$-criterion was computed from the acquired velocity components in order
to identify the vortical structures in the wake (Hunt et al., 1988). This allowed us to visualize the vortices generated at the
tips and roots of the blades in Fig. 13. The figure shows a comparison between the vortices of the baseline case and DIC at
three different stages during the pitch cycle. In the baseline case, we observe how each of the vortices shed by the blades
traveled downstream without much interaction around $x/D = 1$. However, moving further downstream it becomes apparent
that around $x/D = 2$, the vortices started interacting. At this point, the vortices were so close to each other that they started
having a reciprocal influence, causing them to roll up around one another. This process is called leapfrogging, due to the
phenomenon where the first vortex jumps back over the two trailing vortices. The leapfrogging event continued until around



$x/D = 3$ (see black rectangle), where we observe that three vortices merged completely and started to break down afterwards. Leapfrogging plays an important role in the recovery of a wind turbine wake; The wake turbulence induced after the vortex breakdown has been shown to lead to positive entrainment of kinetic energy into the wake (Lignarolo et al., 2014).

Since the breakdown of the tip vortices has a strong influence on the recovery of the wake, it is interesting to look at the difference in the leapfrogging process with DIC. The first time instant from Fig. 13 shows the wake as the pitch angle traveled from its minimum position halfway towards the maximum. Due to the high thrust force associated with this minimum pitch angle, the vortices that were shed are stronger and therefore started interacting at an earlier stage. Here, it is seen that the vortices merged before a distance of $x/D = 2$. As the pitch angle increased and thrust force went down, the strength of the vortices dropped and the location of the leapfrogging event shifted downstream. At the second time instant, we can still see the merging of the vortices at around $x/D = 3$. However, at the third time instant the strength of the vortices has decreased such that the leapfrogging process can no longer be witnessed in the range of measurements. From these visualizations it becomes evident that DIC has a large influence on the breakdown of tip vortices and the accompanying re-energizing of the wake.

## 4 Discussion

The results from the PIV measurements were investigated from three different perspectives: from a time-averaged, periodic and instantaneous point of view. Especially for the time-averaged results it should be recalled that there was a difference in the average thrust coefficient experienced by the turbine during baseline operation and DIC. Hence, the improved velocity fields for DIC do not provide sufficient evidence for the effectiveness of this control approach. However, it should be noted that during the experiment the difference in average thrust levels was smaller than anticipated (see Fig. 5). Due to the changing wind turbine position, small differences in adjacent parts of the velocity fields were observed. This is particularly the case when we examine the difference in stream-wise velocity at $x/D = 3$ in Fig. 7.

The time-averaged data did show increased TI levels towards the center of the wake as a result of applying DIC (see Fig. 8). Since atmospheric turbulence is a key driver for wake recovery, we believe that the additional turbulence induced by DIC stimulates wake mixing, leading to a higher energy content. The TI contours also made it apparent that the influence of the wind turbine's tower and nacelle on the wake was significant and not exclusive to the lower half of the wake. We observe how high turbulence levels behind the tower were sucked upward to around hub height due to the rotation of the wake. This observation corroborates findings from high-fidelity simulations (Santoni et al., 2017) and field measurements (Abraham et al., 2019), which showed that both tower and nacelle play an important role in the recovery of the wake, especially in the far wake. The results presented in this paper not only visualize these effects, but also show that the turbulence caused by the tower and nacelle is enhanced by DIC. We believe this to be an important observation, one that should be taken into account in future studies into axial induction control. This holds especially for high-fidelity simulation studies, which often omit the nacelle and tower structures for practical reasons.

Periodically averaged velocity fields were used to determine the hypothetical available power for a second turbine at $x/D = 3$. This measurement was subsequently averaged over time in order to obtain the total power of an array with a second

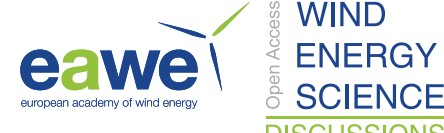



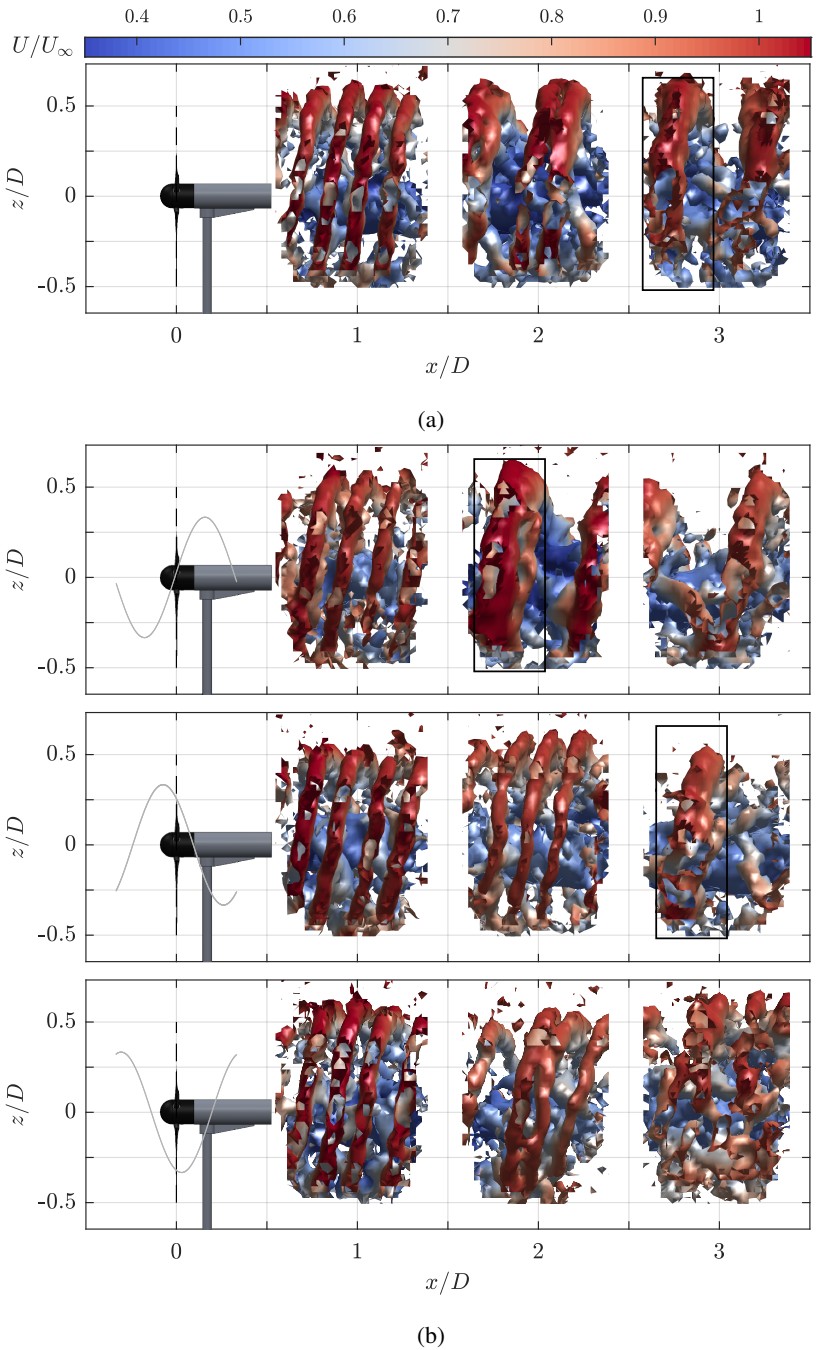

**Figure 13.** Isosurfaces of the instantaneous normalized Q-criterion ($QD^2/U_\infty^2 = 0.1$) visualizing the blade tip vortices for baseline operation (a) and DIC ($St = 0.19$) (b). The DIC case is shown at time instants $t = 0, 0.26$ and $0.52$ s, with the grey line indicates the current pitch angle. The colours of the isosurfaces correspond to the normalized stream-wise velocity component.



hypothetical turbine. The difference in the total power turned out to be minor and in favor of the baseline operation. This was partially due to the relatively high power loss (approximately 10 %) of the model wind turbine, when it was pitching. We expect the power loss to be lower for large-scale wind turbines, which have power curves that are less steep. Furthermore, the loss can be partially ascribed to the $f_{\mathrm{AP}}$ of the baseline control case at $x/D = 3$, which was above the expected trend of the

AP obtained from the measurements at $x/D = 1$ and 2. As mentioned before, we believe this trend break was a result of the varying average thrust experienced at the different measurement positions.

Based on the phase-averaged velocity fields we can conclude that the stream-wise velocity increased when a periodic forcing component was applied to the flow. Furthermore, the results presented in Table 1 show that the increase in mass flow or energy was higher than expected based on the average thrust levels. The exact cause behind the increased energy levels in the wake

is difficult to pinpoint. However, simulations and experiments have indicated that wake recovery is related to the location of the blade tip vortex breakdown (Lignarolo et al., 2014, 2015; Marten et al., 2020). From the instantaneous measurements we can observe that the vortex breakdown for the baseline case occurred after a location of around $x/D = 3$. When applying DIC, we see how the strength of the vortices was affected directly. The stronger vortices accelerated the instability that leads to leapfrogging event and the subsequent vortex breakdown. As a result, we see that the location of the vortex breakdown

changed over time, ranging from $x/D < 2$ to $x/D > 3$. Based on these observations we believe that the wake recovery starts at an earlier stage at some time instants and is spread out over a larger part of the wake when DIC is used. In contrast, the wake starts recovering at a fixed location for the baseline case, achieving a lower overall recovery.

Finally, we will also use this section to briefly discuss some of the challenges that were encountered during the experiment. Our desire for large-scale PIV measurements meant a turbine of limited size, while also being capable of exciting the wake

through a pitch action of the blades. The turbine used in this experiment was able to meet this desire, but due to limitations of the pitch actuators it was not possible to measure at inflow wind speeds $U_\infty > 4\,\mathrm{m\,s^{-1}}$. Operating the open jet at this velocity introduces some unsteadiness to the flow, and the true wind speed measurement at the outlet of the jet is susceptible to measurement errors. The large measurement volume also required the HFSB seeding rake to be positioned in front of the jet in order to cover a large volume with flow tracers. Based on the measurements from Fig. 5, we believe the effect of the seeding

rake is not negligible. Not only did the measured thrust levels differ from the experimentally determined thrust coefficient seen in Fig. 2, they also varied depending on the measurement position of the turbine. For the sake of consistency across measurements, future experiments would do better by operating the turbine at a fixed location, while moving the PIV setup to cover different parts of the wake.

Additional challenges due to the scale of the experiment were encountered during post-processing. The volume contained

a very high number of particles, making it harder for the Shake-The-Box algorithm to reconstruct individual particle tracks. Furthermore, some areas of the wake suffered of having a low particle density. Consequently, we were forced to increase the bin size for the spatial averaging process, resulting in a loss of resolution. Even so, this did not prevent gaps appearing in the instantaneous flow fields. Several methods can be used to improve the resolution using the existing data, ranging from linear interpolation to more complex methods. Schneiders and Scarano (2016) presented an extension to solenoidal filtering (Azijli

and Dwight, 2015), where they also used the velocity derivatives of the particle data. The additional temporal information of the





particles is then used to increase the spatial resolution. Alternatively, statistical methods such as Gaussian process regression can be used to improve the resolution, as such a method can infer the *true* velocity components at any desired location in the measurement volume (de Baar et al., 2014).

## 5  Conclusions

In this paper we investigated the wake of a small scale wind turbine ($D = 0.58\,\mathrm{m}$) operating with dynamic induction control. Measurements of the wake, ranging from $0.5$–$3.5\,D$ downstream, were taken using particle image velocimetry with helium-filled soap bubbles. The results with DIC were compared with those from a baseline controller, which aimed to optimize the turbine power output. Three different pitching frequencies were used for DIC, coinciding with Strouhal numbers of $St = 0.19$, $0.24$, and $0.29$.

Time-averaged measurements showed a decrease in the wake velocity deficit when we applied DIC, which was partially due to a lower average thrust force. The periodic excitation also resulted in higher turbulence levels towards the center of the wake compared to baseline control, indicating enhanced wake recovery. By averaging the velocity data over a single pitch cycle, it was observed how a low velocity region traveling downstream caused the wake to expand and contract locally. The available power $f_{\mathrm{AP}}$ of a hypothetical turbine downstream was subsequently computed using these velocity fields, showing

minor differences in power for the different Strouhal numbers. The measurements were also compared against one dimensional actuator disk theory. Where the latter showed similar levels of $f_{\mathrm{AP}}$ between baseline control and DIC, a clear increase in available power with DIC was observed from the measurements. We believe this to be an additional indication of the enhanced wake recovery due to the periodic excitation.

Finally, the development of blade tip vortices was visualized using the Q-criterion. These figures identified the locations

of the leapfrogging process and subsequent breakdown of vortices, which strongly influence wake mixing and hence the re-energizing of the wake. For the baseline case, we observed a stable location of vortex breakdown around $x/D = 3$. On the other hand, DIC resulted in a continuously changing location depending on the thrust force of the turbine. We believe this to be the main driver behind the enhanced wake recovery of periodic dynamic induction control: The effective wake recovery location is brought forward and expanded over a larger area.

Summarizing, the experiment presented in this paper showed that dynamic induction control increased the wind speed and turbulence intensity in the wake. The periodic thrust force continuously changed the location of vortex breakdown, leading to enhanced wake recovery. The experimental results can be used to improve modelling dynamic excitations of the wake, with the goal of increasing the efficiency of large-scale wind farms.

*Data availability.* Measurement data from the wind tunnel experiment is available upon request.



*Author contributions.*  DvdH prepared and conducted the measurement campaign, analyzed the measurement data, and wrote the paper. JF and MH assisted with the measurement campaign, helped with analyzing the data and with writing parts of the paper. FS aided in the design of the experiment and operation of the measurement equipment. CSF and JWvW supervised the work.

*Competing interests.*  The authors declare that they have no conflict of interest.

*Acknowledgements.*  This work is part of the research programme "Robust closed-loop wake steering for large densely space wind farms"
with project number 17512, which is (partly) financed by the Dutch Research Council (NWO).

Additional thanks goes out to Will van Geest, Alex Rozene Vallespin and Wim Wien for their assistance in preparing the wind turbine model.



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
