# Peer review of "Experimental analysis of the effect of dynamic induction control on a wind turbine wake"

_Wind Energy Science, 2021_

## Referee Comment (RC1)

Review of the paper: "Experimental analysis of the effect of dynamic induction control on a wind turbine wake", by D. van der Hoek et al.

**General comment**

In this paper, an experimental investigation is performed, with the aim of evaluating the impact of dynamic induction control on wake development. The in-wake flow velocity is measured through PIV. Extremely interesting the analysis on the tip vortex behavior. Such an analysis appears to be even more interesting given the fact that it is completely done experimentally: the effort that the authors did to perform such an analysis and to interpret the results is praiseworthy. To my best knowledge, this work represents (unfortunately with some limitations) the first attempt to understand the working principle of dynamic induction control using only experimental data.

I recommended "minor revisions", because there are two concerns that I would like the Authors to address before the final acceptance of the paper. Such issues are reported as "important comments".

**Important comments:**

 Section 2.2: Since the pitch variation employed in the dynamic induction control strategy is not centered around the pitch settings of the greedy control (9 deg), the resulting technique is to be viewed as a combination of both steady and dynamic induction controls. The dynamic induction control is triggered by the periodic pitch variation, while the steady control is due to the steady bias between the baseline pitch and the mean value of the DIC pitch motion.

This can be also viewed in figure 3 in terms of thrust coefficient. Clearly, the DIC is here a combination of steady variation of thrust (one may compare the mean value of the blue line and the value related to the black dash-dotted line), and a periodic variation of the thrust itself, aimed at promoting a faster wake recovery.

This fact may lead to misinterpretation of the results.

Notice that this issue would have been totally avoided, if the baseline control had been implemented with constant pitch equal to about 10.5 deg (i.e., the value associate to the average of the DIC pitch).

Please, explain possible impacts of this inconsistency (it's hard to understand from the presented results whether the impact is negligible or not, at least for the scope of the paper), explain the reason for this choice.

2. Sec 2.3, lines 152. Moving the wind turbine to different positions may imply that the rotor may feel a different flow field (due for example to the fact that the flow modifies within the open chamber, both in terms of velocity and turbulence intensity). Please, give some insight into the possible impact of this issue. Why did the authors decide to keep cameras and led modules fixed and to move the turbine?

Minor comments:

 Section 2.1, line 75-76: Although one may envision that it will be possible to extract the thrust from the tower root bending moment, it could be interesting for readers to have access to some details of the adopted procedure. For example, what is the impact of nodding moment and tower drag on the measurements? How were these effects taken into account?

- 2. Section 2.1, line 83: "calibration function": same issue of the previous point. Some details (or at least a reference, if possible) may be beneficial.
- 3. Section 2.2, It seems that the control is only set for partial power region. If so, this should be noted in the text.
- 4. Sec. 3.1: Where is the wind speed  $U_{\infty}$  measured? This question is connected to the second important comment. There could be a significant difference between the velocity in the different positions at which the turbine is located. This may also explain the difference in the measured thrust coefficient (especially that between x/D=2 and x/D=3, in fig. 5). Moreover, why wasn't the wind speed measured in a previous test with all measurement devices (seeder, cameras, etc.) present, but without the turbine? In this way, one would have had a more reliable velocity measure, at the positions where the turbine would have been installed.
- 5. Sec 3.1, line 183: "we believe that the principles... can still be investigate ...". Here again, many of the incongruencies could have been avoided with a slightly different testing execution. Please, explain the reason why the Authors believe that.
- 6. Fig. 10 and 11, it's a pity that an test with steady induction control with a pitch setting equal to the mean value of the DIC pitch wasn't carried out. One could have had access to what is due to steady induction, and what to dynamic induction.
- 7. Line 287-289: The analysis is extremely interesting; even more, if one considers that it came out of experimentation. However, the contour of the baseline case at 2D is qualitatively similar to that of the DIC case (first plot case b). Why do Authors say "Here, it is seen that the vortices merged before a distance of x/D = 2"?

---

## Author Comment (AC1)

| Date | April 21, 2022 |
| Our reference | n/a |
| Your reference | n/a |
| Contact person | Daan van der Hoek |
| Telephone/fax | +31 (0)15 278 5623 / n/a |
| E-mail | D.C.vanderHoek@tudelft.nl |
| Subject | Response to referees |

**Delft University of Technology**

Delft Center for Systems and Control

Address
Mekelweg 2 (3ME building)
2628 CD Delft
The Netherlands

www.dcsc.tudelft.nl

Anonymous Referee #1, Anonymous Referee #2
*Referees, Wind Energy Science*

Dear Referees,

The authors would like to thank you for all the constructive comments we received on the paper. We believe that the comments have helped us improve the quality of the paper. In our attempt to take your feedback into consideration, some parts of the paper have been revised. In this document, we will try to respond to all the points that were raised by the referees and we will provide a detailed overview of the changes made to the paper. The document consists of two sections where we will address the comments from each of the referees.

Yours sincerely,

Daan van der Hoek
Joeri Frederik
Ming Huang
Fulvio Scarano
Carlos Simao Ferreira
Jan-Willem van Wingerden

| Enclosure(s): | Response to comments of Anonymous Referee #1 |
| | Response to comments of Anonymous Referee #2 |

**Response to comments of Anonymous Referee #1**

Important comments:

- Section 2.2: Since the pitch variation employed in the dynamic induction control strategy is not centered around the pitch settings of the greedy control (9 deg), the resulting technique is to be viewed as a combination of both steady and dynamic induction controls. The dynamic induction control is triggered by the periodic pitch variation, while the steady control is due to the steady bias between the baseline pitch and the mean value of the DIC pitch motion. This can be also viewed in figure 3 in terms of thrust coefficient. Clearly, the DIC is here a combination of steady variation of thrust (one may compare the mean value of the blue line and the value related to the black dash-dotted line), and a periodic variation of the thrust itself, aimed at promoting a faster wake recovery. This fact may lead to misinterpretation of the results. Notice that this issue would have been totally avoided, if the baseline control had been implemented with constant pitch equal to about 10.5 deg (i.e., the value associate to the average of the DIC pitch). Please, explain possible impacts of this inconsistency (it's hard to understand from the presented results whether the impact is negligible or not, at least for the scope of the paper), explain the reason for this choice.

  The authors agree that the point the reviewer addresses is a very important one, and it should be made clear in the paper what this implies for the results that are presented. There are two reasons as to why the pitch angle from baseline case (9 deg.), which was selected to optimize the power production, has a steady offset with respect to DIC (10 deg.). First, the choice for this offset was consistent with the experiments performed by Frederik et al. [1], where the implementation of DIC was also asymmetrical compared to the baseline case. Similar to their experiment, the original plan was to include a second turbine in the wake of the first turbine to measure the effect of DIC directly downstream. Unfortunately, we were not able to perform these measurements due to some issues we experienced with the experiment setup. The second reason is that the current pitch angles used with DIC provide a higher $\Delta C_t$, which is already lower than the thrust variation used in [1]. We were worried that lower thrust variations would make the effects of DIC less visible in the PIV and power measurements. The following lines were added to Sect. 2.2 of the paper:

*"To ensure that a range of different frequencies around this reported optimal frequency could be tested, we varied the blade pitch angle between 8–12°. Note that this variation was not centered around the pitch angle of $\beta = 9°$ that is used to achieve optimal power production. This steady offset in pitch angles between baseline operation and DIC is consistent with the implementation of Frederik et al (2020). Similar to their experiment, the original plan was to include a second turbine in the wake of the first turbine. This would have allowed us to directly measure the effect of DIC on the power of a downstream turbine. Due to some issues we experienced with the experiment setup, this turned out to be infeasible. Another reason for the steady offset in pitch angle was to ensure a higher thrust variation and smaller loss of power according to the aerodynamic coefficients from Fig. 2."*

By doing additional measurements with static induction control (10 deg. pitch), as the reviewer suggests, we could have identified the contributions of the static and dynamic thrust on the wake recovery. Due to the limited amount of time we had access to the testing facilities, we did not manage to do this either. In hindsight, it would have made it easier to interpret the results if we had applied the dynamic pitch signal around the baseline pitch angle of 9 degrees. This will be taken into account for future experiments.

As to the impact of the different average thrust levels on the results, we can look at this from different perspectives. As is already mentioned in the paper, the average velocity deficits presented in Sec. 3.2 do not differentiate between the effects of the steady or dynamic thrust variations. Using a similar average thrust for DIC as the baseline case would probably have resulted in higher velocity deficits than the current ones. For the analysis of the $f_{AP}$ using the periodically averaged velocity fields in Sec. 3.3, we are mostly comparing the control methods at $x/D = 3$. At this location, the average thrust is very similar, meaning the results are only affected slightly by the thrust problem. Furthermore, when analyzing the tip vortex development the average thrust plays a smaller role. We expect it would not have affected the leapfrogging mechanism significantly, apart from moving all locations upstream for a higher average thrust.

The following was added to the paper:

(Sect. 3.3) *"... a hypothetical turbine located at $x/D = 3$ are given in Fig. 12. Note that the average thrust force of the two control methods was almost equal at this location, allowing a more straightforward comparison of their individual results. When comparing the total power coefficient of the baseline case to DIC, we see that ..."*

(Sect. 4) *"... While the difference in average thrust between baseline and DIC makes it difficult to directly compare all measurement results, we believe the analysis of the tip vortices is not affected heavily. If anything, implementing DIC at a higher average thrust would likely have triggered the leapfrogging event at an earlier stage."*

- Sec 2.3, lines 152. Moving the wind turbine to different positions may imply that the rotor may feel a different flow field (due for example to the fact that the flow modifies within the open chamber, both in terms of velocity and turbulence intensity). Please, give some insight into the possible impact of this issue. Why did the authors decide to keep cameras and led modules fixed and to move the turbine? The referee is correct that the properties of the flow change as we move downstream due to the contraction of the jet. Lignarolo et al. [4] investigated the stability of the open jet using hotwire measurements. Based on their analysis, it appears that the inflow velocity for our turbine does not change depending on the measurement position (0.6–2.4 m from the outlet). The turbulence intensity is slightly more sensitive to the distance from the jet, but is still within the range of 2% that is mentioned in Sect. 2.1. So indeed, changing the position probably had a small impact on the measurement results, although we believe the presence of the seeder had a larger effect on the discrepancies between the different measurement positions. The reason for changing the measurement position of the turbine again has to do with the limited amount of time we had access to the wind tunnel facilities. Moving the camera to a new position requires a calibration process which is more time consuming than simply moving the turbine. Nevertheless, we agree that this introduced additional measurement uncertainties, most of all due to the varying distance between seeder and turbine. This will also be taken into account for follow-up experiments in order to limit the amount measurement uncertainties. The following text was added to Sect. 2.3:
  *"Repositioning the turbine instead of the measurement system was done to save time when carrying out the experiments, as this removed the need to calibrate the PIV system when different parts of the wake were measured. Lignarolo et al. (2014) measured the properties of the open jet for different locations downstream, and found that the inflow velocity and turbulence intensity were affected due to the shear layer of the jet. Based on their analysis, we do not expect large deviations in inflow velocity but possibly a minor change in turbulence intensity when moving the turbine from 1–4 D with respect to the outlet of the jet. The presence of the HFSB seeder was expected to have a larger impact on the conditions upstream of the turbine, however, it is unknown how significant these changes were."*

Minor comments:

- Section 2.1, line 75-76: Although one may envision that it will be possible to extract the thrust from the tower root bending moment, it could be interesting for readers to have access to some details of the adopted procedure. For example, what is the impact of nodding moment and tower drag on the measurements? How were these effects taken into account? Some details on the strain gauge calibration process have been added to Sect. 2.1 of the paper:

*"To measure the deformation of the tower, strain gauges were applied at the bottom on the front and back in a full Wheatstone bridge configuration. The strain gauges were calibrated by hanging weights on the back of the nacelle, mimicking the force moment around the tower base originating from the thrust force. This process resulted in a linear expression of the tower moment as a function of the strain gauge voltage. This expression was used to reconstruct the thrust force acting on the rotor from the measured voltage. ...*
*The contribution of the tower and nacelle to the thrust force was measured by removing the blades from the turbine, and was observed to consist of approximately 3% of the total thrust force at the optimal operating settings."*

- Section 2.1, line 83: *"calibration function"*: same issue of the previous point. Some details (or at least a reference, if possible) may be beneficial. Please see the response to the previous comment.

- Section 2.2, It seems that the control is only set for partial power region. If so, this should be noted in the text. The paper has been adjusted accordingly in Sect. 2.2:
  *"Note that during the experiment, the turbine was only operated in the partial load region for which this control law is valid."*

- Sec. 3.1: Where is the wind speed $U_\infty$ measured? This question is connected to the second important comment. There could be a significant difference between the velocity in the different positions at which the turbine is located. This may also explain the difference in the measured thrust coefficient (especially that between x/D=2 and x/D=3, in fig. 5). Moreover, why wasn't the wind speed measured in a previous test with all measurement devices (seeder, cameras, etc.) present, but without the turbine? In this way, one would have had a more reliable velocity measure, at the positions where the turbine would have been installed. The wind tunnel was operated by setting a reference wind speed $U_\infty$, which in turn was automatically regulated using feedback from a built-in pitot tube at the outlet of the open jet. Air density was continuously calibrated using real-time monitored temperature and air pressure. The free stream velocity is then determined using the dynamic pressure and air density measurements. This is standard practice for all experiments carried out at the OJF of TU Delft. As is mentioned in Sect. 3.1, the wind speed at which the wind tunnel was operated made it more susceptible to measurement errors, possibly resulting in small deviations of $U_\infty$ for different measurements.

In an experiment from Jux et al. [3], a similar but slightly smaller seeder was used in front of the measurement object. They reported an increase in turbulence to 1.9 % at 2 m downstream, while the mean flow velocity was not visibly affected. In our experiment, measurements of the flow were also taken without the presence of the turbine, which showed a deviation in $U_\infty$ of approximately 1%, and average turbulence intensity of $5\%$. However, it should be noted that these measurements were only taken at a distance of $4D$ from the seeder. Therefore, we cannot say anything about the difference in inflow conditions the turbine experienced at each of the measurement positions. For this reason, we decided not to include this information in the original submission of the paper. The following has been added to Sect. 3.1 of the paper:

*"... Furthermore, we expect that the influence of the seeding rake changes depending on how close the turbine is positioned from it. Measurements of the wind field without a turbine were taken at a distance of $4\,D$ from the seeder, indicating a largely unaltered mean inflow velocity and an average turbulence intensity of 5 %. Unfortunately, there is no information on the actual inflow conditions at each of the turbine positions. ..."*

- Sec 3.1, line 183: *"we believe that the principles ... can still be investigated ..."* Here again, many of the incongruencies could have been avoided with a slightly different testing execution. Please, explain the reason why the Authors believe that. Please also see our response to the first general comment, which already touched upon this point. Furthermore, the analysis of the available power in Sect. 3.3 mostly compares the measurements taken at $x/D = 3$, where the average thrust levels are very similar. Looking at the leapfrogging mechanism for the DIC case, only the location at which the leapfrogging occurs would have moved upstream for a higher average thrust. We expect the change in mechanism to remain the same. The latter is emphasized at the end of Sect. 3.1:

  *"... However, we believe the principles behind dynamic induction control can still be investigated through these measurements, in particular the effect of DIC on tip vortex development, which we believe to be independent from the average thrust force, that is addressed in Sect. 3.4."*

- Fig. 10 and 11, it's a pity that an test with steady induction control with a pitch setting equal to the mean value of the DIC pitch wasn't carried out. One could have had access to what is due to steady induction, and what to dynamic induction. Please refer to our response to the first general comment where this has been addressed as well.

- Line 287-289: The analysis is extremely interesting; even more, if one considers that it came out of experimentation. However, the contour of the baseline case at 2D is qualitatively similar to that of the DIC case (first plot case b). Why do Authors say *"Here, it is seen that the vortices merged before a distance of $x/D = 2$"*? The authors also believe the analysis to be very relevant, and therefore should be made as clear as possible. Judging from just a single frame of the baseline control case, and with the limitations of the resolution, it is very hard to see where the leapfrogging process starts and ends. In order to improve the current figures, additional post-processing of the data was done using the vortex-in-cell (VIC) method [6, 2]. This method offers a higher spatial resolution of the results by solving an optimization problem that takes velocity and vorticity fields into account when minimizing a cost function. With these results, Fig. 13 was updated into two new figures with the higher resolution data. Figure 13 now shows the different stages of the leapfrogging process for the baseline case, while Fig. 14 does the same for the DIC case. We think that with the new figures, the differences in the leapfrogging process can be seen a lot more clearly.

**Response to comments of Anonymous Referee #2**

- Feel free to ignore this comment if it is not possible, but would a potential comparison be comparing the DIC results, to a change in the baseline controller such that it operates at the same steady Ct as the DIC controller as a supplement used in the paper of comparing to expectations from AD?
  The authors agree that such a comparison would be very helpful in identifying the contributions of both dynamic as well as static induction control to the wake recovery. Unfortunately, such measurements were not taken during the experiment and we have not had the opportunity to repeat the experiment under similar conditions.
- Do the authors have plans to test in a similar way the helix method of DIC?
  We are currently preparing for a similar experiment using the helix method for dynamic induction control. However, this requires a large effort in redesigning the current wind turbine nacelle to replicate the individual pitch control capabilities required for the helix method.

- Page 5: Could you add a little more detail on the definition and meaning of the Strouhal number? Is 0.25 the theoretical value or an empirical selection? If empirical, is there a chance that a value selected using computer simulations might differ from a best choice for wind tunnels / field studies? (Reading ahead I see you do try a few so a little more initial context on this number would be helpful). Is there a theory as to which value should be best and why?

  The optimal Strouhal frequency range as mentioned on this page was derived empirically from large eddy simulations in [5]. The value that was found proved to be the most robust against changes in turbine spacing and turbulence intensity. Furthermore, Strouhal numbers ranging from 0.1–0.4 were evaluated in a wind tunnel experiment by Frederik et al. in [1]. Here, it was found that the peak in the wind farm power curve as a function of Strouhal number was close to 0.25, but not centered exactly around this point. Based on the results in these two papers we have chosen to operate our turbine with Strouhal numbers ranging from 0.19–0.29, as we believe it likely for the optimal Strouhal number to lie within this range. The following text was added to Sect. 2.2 for clarification:

  *"Previous studies have indicated that the optimal pitch frequency is found for a particular range of Strouhal numbers, where the Strouhal number is defined as*

  $$St = \frac{fD}{U}, \tag{1}$$

  *with $f$ the excitation frequency. Munters and Meyers (2018) used large eddy simulations to optimize the Strouhal number for periodic DIC and found a value of $St = 0.25$. Wind tunnel experiments carried out by Frederik et al. (2020) used Strouhal numbers ranging from 0.1–0.4, and observed that the total power peaked for $St$ values between 0.2–0.3."*

- Page 11: How are you defining TI? Oscillation in the streamwise flow? Believe it could be an interesting comparison if making similar plots using stream-wise oscillations and crossstream oscillations.

  A definition of the TI, incorporating all three velocity components, used in Fig. 8 is now given in the paper:

  *"The turbulence intensity is defined relative to the undisturbed inflow velocity $U_\infty$ as*

  $$TI = \frac{\sqrt{\frac{1}{3} \sum_{i=1}^{3} \overline{u_i' u_i'}}}{U_\infty}, \tag{2}$$

  *with*

  $$\overline{u_i' u_i'} = \frac{\sum_{k=1}^{N} \left( u_i(t_k) - \overline{u_i} \right)^2}{N}, \tag{3}$$

  *indicating the variance for each of the three velocity components."*

- Page 11: Fig 8: Would a difference row (as in Fig 7) be useful?
  The authors believe that the differences in TI between the two rows of Fig. 8 are a lot clearer than those of the velocity fields in Fig. 7. For this reason, we have chosen not to include an additional figure in form of a difference row.

- Page 16, section 3.4: If you were to look at similar visualizations as Fig 13 at Strouhal numbers where DIC was not effective, what would you see? Would this change in the leapfrogging behavior not occur? Or would it change in a less beneficial way?
  This is an interesting question, which could provide more insight into the effectiveness of DIC as function of the Strouhal number. We believe large eddy simulations would probably serve such an analysis best, as it provides more flexibility for different settings and gives a higher resolution of the turbine wake. Without awaiting such an analysis, we can try to answer this question by taking the Strouhal number to the extremes on both sides of the optimal range. By considering very low Strouhal numbers close to zero, we will approximate static induction control. The location of the leapfrogging is still expected to change over time, as the strength of the vortices either increases or decreases for different pitch angles. However, this change will happen at a very slow rate, which is why it will probably not improve the average mixing process in the wake. When the Strouhal number is increased to higher values, dynamic inflow is believed to play a more dominant role (especially with the time scales found in a wind tunnel setting). When the blade pitch angle is adjusted to a new setpoint, the change in wake velocity is not immediate but occurs with a certain time delay. By increasing the pitch frequency too much, we arrive at the point where the wake velocity is not able to reach the new equilibrium position before the pitch angle is adjusted again. In other words, the amplitude of the effective thrust experienced by the turbine decreases for higher Strouhal numbers, resulting in smaller variations of the leapfrogging location.

**References**

[1] J. A. Frederik, R. Weber, S. Cacciola, F. Campagnolo, A. Croce, C. Bottasso, and J. W. van Wingerden. Periodic dynamic induction control of wind farms: proving the potential in simulations and wind tunnel experiments. *Wind Energy Science*, 5(1):245–257, 2 2020.

[2] Y. J. . Jeon, J. F. G. . Schneiders, M. . Müller, D. . Michaelis, B. Wieneke, Y. J. Jeon, J. F. G. Schneiders, M. Müller, and D. Michaelis. 4D flow field reconstruction from particle tracks by VIC+ with additional constraints and multigrid approximation. *Proceedings 18th International Symposium on Flow Visualization*, 10 2018.

[3] C. Jux, A. Sciacchitano, J. F. Schneiders, and F. Scarano. Robotic volumetric PIV of a full-scale cyclist. *Experiments in Fluids*, 59(4):1–15, 4 2018.

[4] L. E. Lignarolo, D. Ragni, C. Krishnaswami, Q. Chen, C. J. Simão Ferreira, and G. J. van Bussel. Experimental analysis of the wake of a horizontal-axis wind-turbine model. *Renewable Energy*, 70:31–46, 2014.

[5] W. Munters and J. Meyers. Towards practical dynamic induction control of wind farms: analysis of optimally controlled wind-farm boundary layers and sinusoidal induction control of first-row turbines. *Wind Energy Science*, 3(1):409–425, 6 2018.

[6] J. F. G. Schneiders and F. Scarano. Dense velocity reconstruction from tomographic PTV with material derivatives. *Exp Fluids*, 57:139, 2016.